# Retinal Dystrophy Associated with Homozygous Variants in *NRL*

**DOI:** 10.3390/genes15121594

**Published:** 2024-12-12

**Authors:** Jordi Maggi, James V. M. Hanson, Lisa Kurmann, Samuel Koller, Silke Feil, Christina Gerth-Kahlert, Wolfgang Berger

**Affiliations:** 1Institute of Medical Molecular Genetics, University of Zurich, 8952 Schlieren, Switzerland; maggi@medmolgen.uzh.ch (J.M.); kurmann@medmolgen.uzh.ch (L.K.); koller@medmolgen.uzh.ch (S.K.); feil@medmolgen.uzh.ch (S.F.); 2Department of Ophthalmology, University Hospital Zurich and University of Zurich, 8091 Zurich, Switzerland; james.hanson@usz.ch (J.V.M.H.); christina.gerth-kahlert@usz.ch (C.G.-K.); 3Zurich Center for Integrative Human Physiology (ZIHP), University of Zurich, 8057 Zurich, Switzerland; 4Neuroscience Center Zurich (ZNZ), University and ETH Zurich, 8057 Zurich, Switzerland

**Keywords:** NRL, retinitis pigmentosa, retinal dystrophy, splicing, uniparental disomy

## Abstract

**Background/Objectives**: Neural retina leucine zipper (NRL) is a transcription factor involved in the differentiation of rod photoreceptors. Pathogenic variants in the gene encoding NRL have been associated with autosomal dominant retinitis pigmentosa and autosomal recessive clumped pigmentary retinal degeneration. Only a dozen unrelated families affected by recessive *NRL*-related retinal dystrophy have been described. The purpose of this study was to expand the genotypic spectrum of this disease by reporting clinical and genetic findings of two unrelated families. **Methods**: Index patients affected by retinal dystrophy were genetically tested by whole-exome sequencing (WES) and whole-genome sequencing (WGS). Segregation analysis within the families was performed for candidate variants. A minigene assay was performed to functionally characterize a variant suspected to affect splicing. **Results**: Variant filtering revealed homozygous *NRL* variants in both families. The variant in patient A was a small deletion encompassing the donor splice site of exon 1 of transcript NM_006177.3. The minigene assay revealed that this variant led to two aberrant transcripts that used alternative cryptic donor splice sites located in intron 1. In patient B, a stop-gain variant was identified in the last exon of *NRL* in a homozygous state due to maternal uniparental disomy of chromosome 14. **Conclusions**: Our study expands the genotypic spectrum of autosomal recessive *NRL*-related retinal dystrophy. Moreover, it underscores the importance of actively maintaining bioinformatic pipelines for variant detection and the utility of minigene assays in functionally characterizing candidate splicing variants.

## 1. Introduction

The human retina is an organized multilayered tissue structure composed of several cell types, including one glial type (Müller glia) and five major neuronal types (photoreceptors, bipolar cells, retinal ganglion cells, horizontal cells, and amacrine cells) [1]. Topologically, the retina is organized into distinct concentric regions with characteristic photoreceptor subtype compositions and densities [2,3]. Photoreceptors can be divided into rods and cones based on their morphology and light sensitivity. Specifically, cones are responsible for color vision by responding to bright light, whilst rods mediate low-light vision [4].

Development of the human retina is a complex and lengthy process, with rods differentiation finalizing postnatally [1,5]. Photoreceptors arise from retinal progenitor cells (RPCs) that become photoreceptor precursors to finally differentiate into a rod or a cone [6,7]. The differentiation from photoreceptor precursor to a mature rod or cone subtype involves at least six key transcription factors: OTX2, CRX, NRL, NR2E3, RORB, and THRB [7].

Neural retina leucine zipper (NRL) is a key transcription factor in the differentiation of rod photoreceptors [7,8,9]. NRL has been described to directly repress the expression of cone-specific genes (i.e., *THRB* and *OPN1SW*) and induce the expression of rod-specific genes (i.e., *NR2E3*) [1,10,11]. Retinas of Nrl-null mice have no rods and a surplus of S-cones [7,8,9,12,13]. Similarly, genome-edited embryonic stem cells lacking *NRL* differentiate into rod-deprived retinal organoids [9]. Conversely, ectopic expression of Nrl in mice photoreceptor precursors leads to a retina composed exclusively of rods [11].

The *NRL* gene is located on chromosome 14 and has been linked to inherited retinal dystrophies (IRDs) [14,15,16]. Gain-of-function variants at amino acid positions 49, 50, 51, and 96 have been shown to reduce NRL phosphorylation and/or increase rhodopsin promoter activation, leading to autosomal dominant retinitis pigmentosa (adRP, OMIM #613750) [17,18,19,20]. Conversely, loss-of-function variants in *NRL* have been found in patients affected by autosomal recessive clumped pigmentary retinal degeneration, which resembles the clinical phenotype caused by autosomal recessive variants in *NR2E3*, also known as enhanced S-cone syndrome (ESCS, OMIM #268100) [16,21,22,23,24].

To our knowledge, only a dozen unrelated families have been reported with recessive *NRL*-related retinal dystrophy to date [16,21,22,23,24,25,26,27]. In this study, we present two additional unrelated families and expand the genotypic spectrum of recessive *NRL*-related retinal dystrophy. Moreover, this report highlights the importance of keeping bioinformatic workflows for variant detection up-to-date. Finally, one of the variants was functionally characterized by a minigene assay, which revealed variant-induced aberrant splicing events.

## 2. Materials and Methods

### 2.1. Clinical Examinations

In addition to standard ophthalmological examinations and tests such as measurement of visual acuity (VA), slit-lamp anterior eye examination, and dilated fundoscopy, the following clinical data were acquired:

The full-field electroretinogram (ERG) was measured using a Espion system (Diagnosys LLC, Lowell, MA, USA) and Dawson–Trick–Litzkow (DTL; Diagnosys LLC) electrodes positioned at the lower lid margin according to contemporary recommendations of the International Society for Clinical Electrophysiology of Vision (ISCEV) [28,29]. Periods of dark and light adaptation were 20 and 10 min, respectively, and sampling rate was 2 kHz.

Optical coherence tomography (OCT) and fundus autofluorescence (FAF) imaging were performed with the Spectralis OCT device (Heidelberg Engineering GmbH, Heidelberg, Germany). A pragmatic approach was necessary, with the scan settings employed determined at each visit by the age and compliance of the subjects at the time of examination, and the presence/absence of nystagmus.

Kinetic perimetry was performed with an Octopus 900 (Haag-Streit AG, Köniz, Switzerland). As with the OCT, a pragmatic approach was imposed by the age and compliance of the subjects at examination, but at least two kinetic isopters were assessed at each exam.

Retinal imaging (including fundus autofluorescence, FAF) was performed with an Optomap P2000DTx wide-field imaging device (Optos plc, Dunfermline, UK).

### 2.2. Genetic Testing

Genomic DNA (gDNA) was extracted from whole blood in duplicate with the automated Chemagic MSM I system according to the manufacturer’s specifications (PerkinElmer Chemagen Technologie GmbH, Baesweiler, Germany). Genetic testing strategies included whole exome sequencing (WES) and whole genome sequencing (WGS) when WES analysis was inconclusive.

Whole exome sequencing was performed as previously described [30]. Briefly, library preparation was carried out with IDT’s Exome kit v2 (Integrated DNA Technologies, Coralville, IA, USA), and libraries were sequenced on a NextSeq instrument (Illumina, San Diego, CA, USA), according to the manufacturer’s instructions.

Whole genome sequencing was performed as previously described [31]. Briefly, the TruSeq DNA PCR-Free kit (Illumina, San Diego, CA, USA) was used for library preparation, which was sequenced on a NovaSeq 6000 instrument (Illumina, San Diego, CA, USA).

Raw sequencing data were mapped to the Human reference genome hg19 and variants were called as previously described [31]. The analysis pipeline can be found on github (https://github.com/jordimaggi/WGS_analysis_workflow, accessed on 1 August 2024). Only variants within previously described IRD-associated loci were considered (Appendix A). Variants were prioritized based on gnomAD frequencies [32], ClinVar entries [33], phyloP scores [34], CADD v1.6 scores [35], spliceAI scores [36], primateAI scores [37], revel scores [38], sift predictions [39], polyPhen predictions [40], family history, and in-house frequencies [31]. Candidate variants were classified according to ACMG guidelines [41] on the Franklin platform (https://franklin.genoox.com/clinical-db/home, accessed on 12 September 2024), and the HGMD Professional v.2024.2 [42] and LOVD [43] databases were queried for corresponding existing entries.

### 2.3. Segregation Analysis

Segregation analysis for candidate variants was performed for available family members. For this purpose, initial PCR amplification and Sanger sequencing were performed as previously described [30].

### 2.4. Splicing Assay

A minigene assay was performed to functionally characterize a variant in *NRL* that may affect splicing (NM_006177.3:c.-41_-28+23del). The genomic region corresponding to the entire coding sequence of the *NRL* transcript NM_006177.3 (hg19, chr14:24549110-24553922) was amplified by PCR with Phusion High-Fidelity DNA Polymerase (New England Biolabs, Ipswich, MA, USA) from the mother’s gDNA and cloned into the pcDNA3.1 backbone (Invitrogen, Carlsbad, CA, USA) using the Takara In-Fusion HD cloning kit (Takara, Kusatsu, Japan), according to the manufacturer’s instructions. The genotype of the region of interest was verified by Sanger sequencing, as previously described [30].

A reference minigene (corresponding to the reference sequence over the variant region) and a variant minigene (corresponding to the variant sequence) plasmids were transfected into HEK293T cells by Xfect Transfection Reagent (Takara, Kusatsu, Japan), according to the manufacturer’s instructions. Total RNA was isolated from the cells 24 h after transfection and reverse transcribed into cDNA with the NucleoSpin RNA Plus (Macherey-Nagel, Düren, Germany) and SuperScript III First-Strand Synthesis SuperMix (Invitrogen, Waltham, MA, USA) kits, according to the manufacturer’s instructions.

Minigene-derived transcripts were amplified from cDNA by PCR using primers that bind to *NRL* (NM_006177.3) exon 1 and 2, including adapter sequences for the Nanopore PCR Barcoding Kit SQK-PBK004 (TTTCTGTTGGTGCTGATATTGC-forward primer sequence, and ACTTGCCTGTCGCTCTATCTTC-reverse primer sequence; Oxford Nanopore Technologies, Oxford, UK). The PCR was performed according to the Phusion High-Fidelity DNA Polymerase protocol (New England Biolabs, Ipswich, MA, USA) in 50 µL volume with the GC-Buffer and 100 ng of cDNA. Agarose (1%) gel electrophoresis was performed to verify PCR products, which were then purified with AMPure XP beads (Beckman Coulter Life Sciences, Indianapolis, IN, USA) with a 1:1.5 (PCR/bead) ratio and eluted in 50 µL of 1X Tris-EDTA (TE) buffer (Integrated DNA Technologies, Coralville, IA, USA), according to the manufacturer’s instructions. The QuBit dsDNA High-Sensitivity Assay kit (Thermofisher Scientific, Waltham, MA, USA) was used to measure concentration of the purified PCRs.

The Nanopore indexing PCR was performed with 24 µL of purified PCRs at a concentration of 10 ng/µL, 25 µL of Long Amp Taq 2X Master Mix (New England Biolabs, Ipswich, MA, USA), and 1 µL of barcoded universal primers with rapid attachment chemistry from the Nanopore PCR Barcoding kit SQK-PBK004 (Oxford Nanopore Technologies, Oxford, UK), according to the manufacturer’s instructions. After purification with AMPure XP beads with a 1:1 (PCR/bead) ratio, concentration and size distribution of the PCR products were measured with QuBit dsDNA High-Sensitivity Assay kit and a Bioanalyzer High-Sensitivity DNA kit on a Bioanalyzer 2100 instrument (Agilent Technologies, Santa Clara, CA, USA). The purified PCRs were pooled to a total of 70 fmol in a final volume of 10 µL, and the rapid 1D sequencing adapters were attached by adding 1 µL of RAP and incubating the reaction mix for 5 min at room temperature.

The final library was sequenced with an R9.4.1 (FLO-MIN106D) Flow Cell on a MinION Mk1C instrument (Oxford Nanopore Technologies, Oxford, UK) using the MinKNOW v.23.07.5 software, according to the manufacturer’s instructions. The wf-basecalling v.1.0.1 workflow on the EPI2ME v.5.1.3 platform (Oxford Nanopore Technologies, Oxford, UK) was used to convert pod5 files to FASTQ files, which were demultiplexed with the Barcoding Analysis module on MinKNOW v.23.07.5 software. Alignment to the minigene construct sequence was performed with minimap2 v.2.26 using the “splice” option [44]. The alignment file was sorted, indexed, and converted to BAM with samtools v.1.18 [45].

Nanopore alignment files were used to identify and quantify full-length transcripts as previously described [46]. Briefly, the JWR_checker.py script from NanoSplicer v.1.0 [47] detected splice junctions from the alignment files. High-quality reads (transcripts) were isolated and quantified from the output file using the script from our previous study [46].

## 3. Results

### 3.1. Clinical Findings

#### 3.1.1. Patient A

Background and clinical examination: the female patient of Middle Eastern origin was referred to our clinic aged 7 years for a second opinion regarding decompensating exophoria or intermittent strabismus in the right eye. The patient was reported as being more clumsy than her younger sister, but no nyctalopia or other visual symptoms or family history of ocular disease were reported. Consanguinity was present, with her parents being first cousins and the parents of her maternal grandfather also being cousins. Refractive error (mean spherical equivalent [MSE] in diopter’s right eye +3.25, left eye +3.75) was corrected with spectacles. VA at referral was 1.0 Snellen decimal in both eyes. Fundoscopy revealed a normal optic disc, macula wrinkling, reduced macular reflex, clumped hypopigmented patches at the level of the retinal pigment epithelium (RPE) along the temporal vascular arcades, hypo- and hyperpigmented RPE but no bone spicule pigmentation at her first visit. At the age of 16 years, pigmentary changes and atrophic areas had increased and were more pronounced relative to age 7, whereas macula wrinkling was reduced. The atrophic areas corresponded to hypofluorescent regions visible in the AF images. Functional and imaging findings are shown in Figure 1.

ERG: at age 7, rod responses were non-recordable in both eyes. Both dark-adapted mixed rod–cone and light-adapted cone responses to single flash stimuli exhibited an electronegative ERG configuration, with normal or slightly reduced a-wave amplitudes and b-waves of lesser amplitude than the a-waves. Cone-mediated flicker responses were affected by blink artifacts due to photophobia in both eyes; the results were not interpretable in the right eye and most likely reduced and delayed in the left eye. At age 12, cooperation was improved and the ERG results had slightly worsened. The negative ERG configurations to dark-adapted and light-adapted single flash stimuli remained, whilst the flicker responses were again not interpretable due to blink artifacts.

OCT: central retinal thickness remained stable over a period of 8 years, with no schisis, cystoid lesions, or edema visible at any time. Patches of outer retinal atrophy at the temporal arcades were visible and also remained stable over the follow-up period.

Visual fields: at age 7, all isopters were concentrically reduced, remaining approximately stable up to the most recent test at age 15.

#### 3.1.2. Patient B

Background and clinical examination: A male patient of Swiss origin was first referred to our clinic at age 7 for investigation of a suspected chorioretinopathy. At 3 months old, he developed nystagmus and had an isolated fit; electroencephalography and magnetic resonance imaging were performed at this time and found to be normal. He was diagnosed with growth hormone deficiency, clumsiness, and slight developmental delay. Maternal uniparental disomy of chromosome 14 was identified at that time. No consanguinity or family history of ocular disease was present. Examination at age 7 years confirmed horizontal pendular nystagmus and hyperopic astigmatism in both eyes. Habitual visual acuity at this examination was 0.3 Snellen decimal in both eyes. Fundus examination revealed normal optic discs, round and oval clumped hypopigmented patches along the temporal vascular arcades and midperipheral retinal atrophic changes in both eyes and also preretinal fibrosis in the temporal midperiphery in the right eye. Functional and imaging findings are shown in Figure 2.

ERG: at age 13, the ERG in the right eye was not analyzable due to nystagmus and consequent eye movement artifacts. In the left eye, no rod responses were recordable; mixed rod–cone responses were reduced and delayed and exhibited an electronegative ERG configuration. The cone single flash response was reduced and delayed and had a low b-/a-wave amplitude ratio of 1.05 but was not conclusively electronegative or pseudo-negative in configuration. The flicker response in the left eye was not interpretable.

OCT: nystagmus precluded the acquisition of volume scans; however, it was possible to acquire single b-scans of low quality in both eyes. These revealed the presence of grade 1 foveal hypoplasia [48] and the presence of schisis and cystoid lesions temporal to the fovea.

Visual fields: at age 8, it was only possible to measure I/4E isopters due to the age of the patient, and the visual field was circumferentially reduced to approximately 20° OD, 30° OS. By age 11, a more reliable examination was possible, and the results were stable or improved relative to age 9. Two further examinations at age 13 confirmed broadly intact isopters to larger stimuli but constricted isopters to medium and smaller stimuli. In summary, perimetry was abnormal in both eyes, but a reliable judgment with regard to progression was not possible.

### 3.2. Identification of Homozygous Candidate Pathogenic Variants in NRL

#### 3.2.1. Patient A

In patient A, the filtering strategy applied to the WES data revealed three candidate variants in *SAMD11*, *HKDC1*, and *TRNT1* (Table 1). All variants were heterozygous and were found in genes described to cause autosomal recessive forms of RP [49,50,51]. For this reason, we proceeded with WGS as the second-tier assay. WGS data revealed the presence of a small homozygous deletion (37 bp; NM_006177.3:c.-41_-28+23del; Table 1) overlapping with the exon-intron boundary of noncoding exon 1 of *NRL* transcript NM_006177.3 (equivalent to exon 2 of transcript NM_006177.5). The variant has no frequency in the gnomAD database. The IDT’s exome kit v2 panel captures this exon, and the variant should have been detected during our WES analysis. However, the variant was not called by GATK’s variant caller *UnifiedGenotyper*, which was part of the bioinformatic pipeline at the time of the original analysis. Retrospectively, applying the more recent *HaplotypeCaller* algorithm would have identified the variant.

This *NRL* variant abolishes the exonic donor splice site (Appendix A); therefore, it was suspected to affect splicing. The affected exon is part of the 5′ UTR and is not included in all *NRL* transcripts; in fact, it is exclusively integrated in transcript NM_006177 (newest version NM_006177.5). The human retinal transcriptome (https://oculargenomics.meei.harvard.edu/retinal-transcriptome/, accessed on 21 October 2024) [52] and RNA-seq data of retinal organoids from our previous studies [53,54] provide evidence that this transcript may be the most abundant *NRL* splice isoform in the retina (Appendix A). Segregation analysis confirmed that the variant segregated with the disease within the family (Figure 3).

#### 3.2.2. Patient B

In patient B, three heterozygous candidate variants in *USH1C*, *IFT27,* and *ADGRA3* and one homozygous variant in *NRL* were found after filtering the WES data (Table 2). While pathogenic variants in *USH1C* and *IFT27* are associated with autosomal recessive Usher syndrome type 1C and Bardet–Biedl syndrome 19, respectively [55,56], mutated *ADGRA3* is linked to autosomal recessive RP [57].

The homozygous *NRL* variant is located in exon 3 of the transcript and leads to a premature termination codon at amino acid position 182 (NM_006177.3:c.544C>T; p.(Gln182*)). No frequency is listed in the gnomAD database for this variant in version v2.1.1; the newest version (v.4.1.0) reports an overall frequency of 0.00082% (11/1′347′978 alleles). Segregation analysis revealed that the mother, but not the father, of patient B is a carrier of the variant (Figure 4). Therefore, the previously diagnosed uniparental (maternal) disomy is the most likely cause for homozygosity of this rare sequence variant in the patient.

### 3.3. Functional Analysis of a Novel Small Deletion in NRL

A minigene construct based on the pcDNA3.1 backbone was generated to functionally characterize the *NRL* variant NM_006177.3:c.-41_-28+23del. The 4813 bp region corresponding to the exons and introns of transcript NM_006177.3 was cloned into the backbone (refer to the Materials and Methods Section 2.4). The reference sequence minigene should result in the expression of a “WT” transcript corresponding to exons 1, 2, and 3 of the *NRL* transcript NM_006177.3 (total length of 1947 bp). The primers used for the amplification of the minigene transcripts were designed to bind to exons 1 and 2 for a PCR product of 388 bp (432 bp, including the Nanopore Barcoding Kit overhangs) for the WT transcript. The gel electrophoresis revealed the presence of multiple PCR products (Appendix A).

The expected WT transcript (T1; Figure 5 and Table 3) was identified by Nanopore sequencing in both reference and variant minigene assays at different levels. In the reference minigene assay, the WT transcript represented 99.6% of the reads. Conversely, the variant minigene assay highlighted 34.5% of residual WT transcript.

In the variant minigene assay, two additional transcripts were identified (T2 and T3; Figure 5 and Table 3). Transcript T2 utilizes a cryptic donor splice site located at position c.-28+102 (hg19 chr14:g.24553627) instead of the deleted natural donor splice site and was found to be represented by 23.3% of the reads (453 bp including the overhangs). Similarly, transcript T3 uses another cryptic donor splice site located further downstream at position c.-28+596 (hg19 chr14:g.24553133), and its abundance was measured at 10.9% (947 bp including overhangs).

The cryptic donor splice site in T2 is very weak, according to prediction tools on the Alamut Visual Plus software v.1.6.1 (Appendix A). Conversely, T3 is characterized by the use of a strong cryptic donor splice site (Appendix A). Additionally, gel electrophoresis (Appendix A) seemed to indicate the longer PCR product (representing transcript T3) to be more abundant than the other transcripts. These observations are not consistent with the quantifications extracted from the Nanopore sequencing datasets, which found T2 to be more abundant than T3.

The detection of WT transcript in the variant minigene assay was an unexpected finding as its expression cannot result in WT transcript due to the lack of the natural exon 1 donor splice site. The presence of the WT transcript in the sequencing results may be due to endogenous expression in HEK293T cells; in this case, the WT transcript would be preferentially amplified during the PCR. Another source of “transcript contamination” could be index hopping, although the relatively high abundance of these reads makes it less likely.

## 4. Discussion

We report two unrelated patients affected by autosomal recessive retinal dystrophy who carry homozygous pathogenic variants in *NRL*.

Both of our patients exhibited early and severe retinal dystrophy, which appeared to be progressive over the follow-up period of 6 to 9 years. The retinal changes with a clumped pattern at the level of the RPE, mainly along the temporal vascular arcades, are typical for *NRL*- or *NR2E3*-related retinal dystrophy [24]. However, this characteristic pattern of fundus changes is less obvious at early ages. Thus, the diagnosis may be delayed if electrodiagnostic tests are difficult to perform and genetic evaluation is not available at first. We were not able to ascertain whether our patients also exhibited the enhanced S-cone function documented by other authors in cases of homozygous *NRL* variants [16,23,24]. Specific testing of S-cone function is not included in the standard ERG recording protocol routinely used in our clinic. As S-cone ERG recording can be demanding for patients, it is, in any case, uncertain if usable measurements could have been recorded in our younger patients (particularly patient B, whose nystagmus caused considerable artifacts when recording the standard ERG). However, the standard ERG findings described in some patients with homozygous *NRL* variants and enhanced S-cone responses appear partly distinct from those in our patients, with Iarossi et al. documenting residual rod function and reduced but normally configured (i.e., not electronegative) responses [24]. In contrast, the patients described by Newman et al. both had non-recordable rod responses, and one had a negative ERG [23], similar to both of our patients. Additionally, Newman’s patient with a negative ERG had a fundus appearance similar to both of our patients.

Of note, both of our patients were asymptomatic at first examination, with their visual impairment being noticed by family members. Now in their teenage years, they both struggle to accept that their visual function is not sufficient to hold a driving license. The asymptomatic nature of their retinal dystrophies is likely to be consistent with an onset in early infancy. Other authors have described patients with Ser50Thr mutations in *NRL* and adRP as exhibiting night blindness symptoms from birth to 16 years and subjective loss of peripheral vision between ages 20–37 years [20]. In comparison, our patients exhibited severe but previously described fundus changes [24] (distinct from those typically seen in adRP) at a younger age, with ERG findings likely less severe.

The variant in patient A (NM_006177.3:c.-41_-28+23del) is a deletion that disrupts the natural donor splice site of exon 1 of transcript NM_006177.3 (or exon 2 of transcript NM_006177.5), which encodes part of the 5′ UTR region. This particular exon is not part of the MANE Select transcript (NM_001354768.3); however, RNA-seq data suggest that this exon is included in the vast majority of transcripts in the retina. The variant was not detected during WES analysis due to an outdated bioinformatic pipeline. A minigene assay revealed two transcripts characterized by aberrant splicing events in the minigene sequence containing the variant.

The variant in patient B (NM_006177.3:c.544C>T) located in the last exon of the gene leads to a premature stop codon and, thus, probably to a truncated protein. The variant has already been described in the literature as pathogenic in the context of non-syndromic RP [22]. The premature translational arrest may lead to the loss of the leucine zipper domain of NRL, which would impair the function of this transcription factor [22]. Since the mother of patient B is a carrier of the variant and the variant was not found in the father of the patient, homozygosity of this rare sequence variant can be explained by the diagnosed uniparental (maternal) disomy of chromosome 14. If uniparental disomy of a specific chromosome is known, targeted sequencing of loci associated with the phenotype located on that specific chromosome instead of WES has the potential to expedite the identification of the molecular diagnosis.

This study confirms the usefulness of minigene assays for the characterization of aberrant splicing events due to candidate splicing variants. Simultaneously, these assays should be interpreted cautiously. The limited capacity (in terms of genomic context), the reliance on PCR amplification, and the limitations intrinsic to the sequencing technology can introduce important biases in the nature and proportions of detected splicing events. In fact, a discrepancy between the Nanopore sequencing quantifications and the gel electrophoresis of the PCR products for variant NM_006177.3:c.-41_-28+23del was recognized in this study. However, the identification of aberrant splicing events remains a strong indication that the variant affects splicing patterns and would fulfill the PS3 category criterion of the ACMG’s classification system.

The identification and characterization of pathogenic variants responsible for the clinical manifestations in patients affected by inherited retinal dystrophies (IRDs) is crucial to determining their eligibility for targeted therapies. Gene therapies are promising treatment strategies for IRDs. Several gene therapies for different target genes are being tested in clinical trials currently and one has already been commercialized under the name Luxturna for patients with *RPE65*-related retinal dystrophy [58].

## Figures and Tables

**Figure 1 genes-15-01594-f001:**
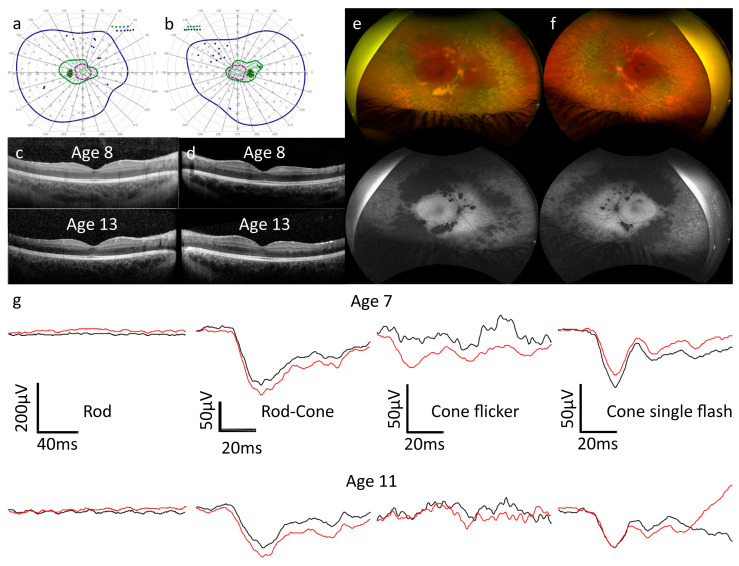
(**a**–**g**). Clinical and imaging data for Patient A. (**a**,**b**) At age 15, kinetic visual field testing revealed concentrically reduced isopters in left (**a**) and right (**b**) eyes. V4e, I4e, and I2e isopters and static points are displayed in dark blue, dark green, and purple, respectively. Open circles represent stimuli presented statically that were seen by the patient, whilst filled circles represent static stimuli that were not seen by the patient. I1e stimuli were also presented, but the isopter is constricted and not visible on the diagram. (**c**,**d**) Optical coherence tomography findings were unchanged in the right (**c**) and left (**d**) eyes between the ages of 8 and 13. (**e**,**f**) Optos wide-field imaging at age 16 revealed clumped hypopigmentation along the vascular arcades and peripheral retinal atrophy corresponding to the autofluorescence (AF) pattern, which demonstrated centrally preserved and increased AF in both the right (**e**) and left (**f**) eyes. (**g**) Electroretinography (ERG) at age 7 revealed, from left to right, the following: non-recordable rod responses; reduced and delayed rod–cone responses with electronegative ERG configuration; light-adapted flicker responses contaminated by artifacts and therefore not interpretable; and reduced cone single flash responses with delayed a-waves and electronegative ERG configuration. At age 11, the rod–cone and cone single flashes had worsened, with the electronegative configuration remaining. Black traces are from the right eye and red traces from the left eye.

**Figure 2 genes-15-01594-f002:**
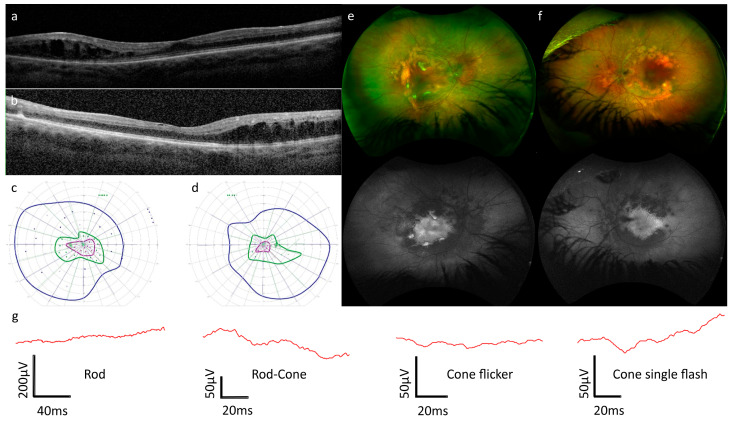
(**a**–**g**). Clinical and imaging data for Patient B. (**a**,**b**) Optical coherence tomography showed grade 1 foveal hypoplasia and schisis with cystoid changes temporal to the fovea in both the right (**a**) and left (**b**) eyes. More detailed scanning was not possible due to nystagmus. (**c**,**d**) Kinetic perimetry at age 13 revealed intact isopters to larger stimuli and constricted isopters to smaller stimuli in both left (**c**) and right (**d**) eyes. V4e, I4e, and I2e isopters and static points are displayed in dark blue, dark green, and purple, respectively. Open circles represent stimuli presented statically that were seen by the patient, whilst filled circles represent static stimuli that were not seen by the patient. (**e**,**f**) Optos wide-field imaging revealed patchy hypopigmentation along the temporal vascular arcades corresponding to the reduced autofluorescence pattern in both right (**e**) and left (**f**) eyes. (**g**) Electroretinography (ERG) at age 13, was challenging due to nystagmus. From left to right: the rod response was non-recordable; the rod–cone response was reduced and delayed, with negative ERG configuration; analysis of the flicker response was not possible due to artifact; the cone single-flash response was reduced and delayed, but due to artifact, a negative ERG configuration could not be conclusively confirmed. All traces are from the left eye only (it was not possible to analyze the responses from the right eye due to nystagmus-related artifacts).

**Figure 3 genes-15-01594-f003:**
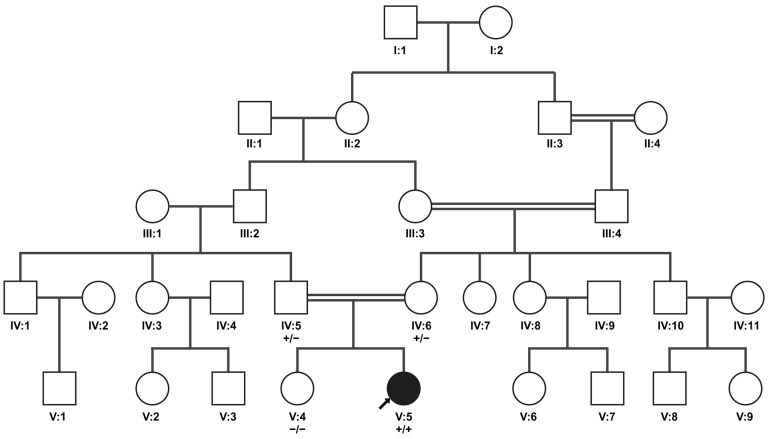
Family pedigree for patient A illustrating cosegregation of *NRL* variant NM_006177.3:c.-41_-28+23del. The arrow indicates patient A (index patient; V:5). The genotype of the family members tested (IV:5, IV:6, V:4, and V:5) is represented underneath their identifiers; plus symbols (+) refer to the variant allele, and minus symbols represent the major allele. The two alleles are separated by a slash symbol. Both parents carry the variant heterozygously, while the sibling does not carry the variant. Double lines indicate consanguinity. Created in BioRender. Maggi, J. (2024) https://BioRender.com/m01g165.

**Figure 4 genes-15-01594-f004:**
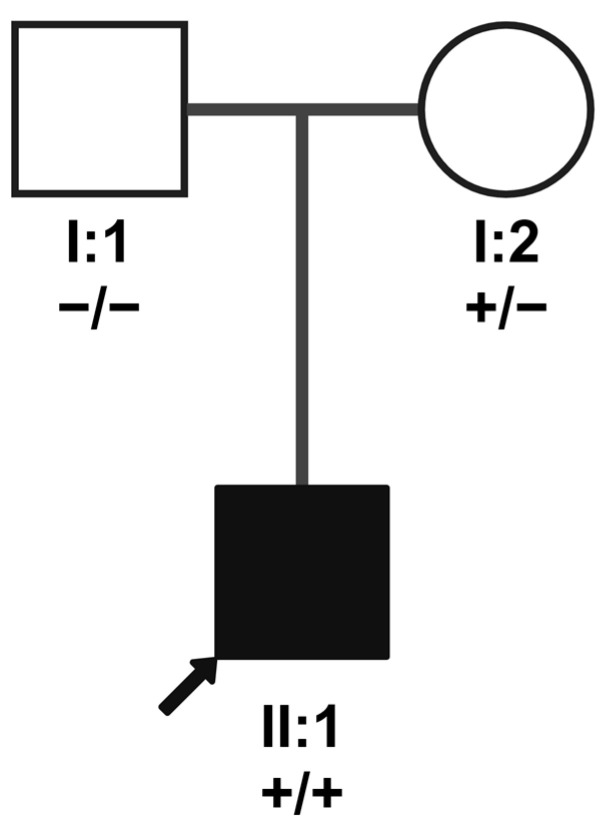
Family pedigree for patient B illustrating segregation of *NRL* variant NM_006177.3:c.544C>T. The arrow indicates patient B (index patient; II:1). The genotype of the parents (I:1 and I:2) is represented underneath their identifiers; plus symbols (+) refer to the variant allele, and minus symbols represent the major allele. The two alleles are separated by a slash symbol. Only the mother carries the variant (heterozygously); maternal uniparental disomy of chromosome 14 had been previously diagnosed in the index patient. Created in BioRender. Maggi, J. (2024) https://BioRender.com/y07r686.

**Figure 5 genes-15-01594-f005:**
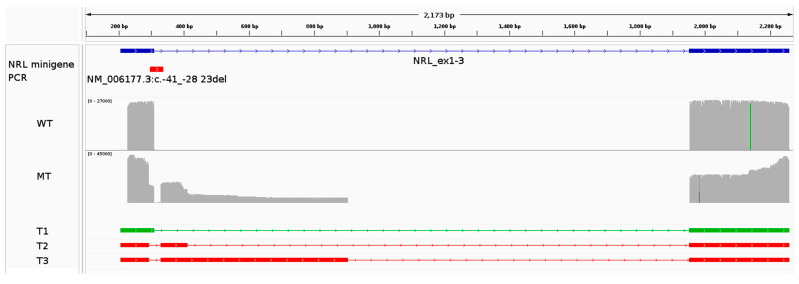
Functional characterization of the *NRL* variant NM_006177.3:c.-41_-28+23del using a minigene assay. The panel shows an IGV screenshot highlighting the construct’s characteristics (displayed in blue), followed by the coverage plots (grey) for the reference (WT) and variant (MT) minigenes. Mismatches between the sequence of the reads and the reference sequence are highlighted in the coverage plot by a color different than grey. An overview of each transcript (name T#) identified in the analysis can be seen underneath the coverage plots. The green transcript represents the expected major (WT) transcript.

**Table 1 genes-15-01594-t001:** Candidate variants for patient A detected in WES and WGS datasets. Classification according to American College of Medical Genetics and Genomics (ACMG) guidelines from the Franklin platform.

Gene	cNomen	Zyg.	gnomAD All (%)	ACMG	LOVD	ClinVar	HGMD	Testing Assay
*SAMD11*	NM_152486.4:c.682_683insT	Het.	0.032	VUS	-	VUS	-	WES
*HKDC1*	NM_025130.4:c.1588G>A	Het.	0.002	VUS	-	-	-	WES
*TRNT1*	NM_182916.2:c.43C>T	Het.	0.007	VUS	VUS	VUS	-	WES
*NRL*	NM_006177.3:c.-41_-28+23del	Hom.	NA	VUS	-	-	-	WGS

Abbreviations: cNomen, Human Genome Variation Society (HGVS) cDNA-level nucleotide change nomenclature; gnomAD all (%), genome aggregation database v.4.1.0 overall minor allele frequency in percentage; ACMG, American College of Medical Genetics and Genomics guidelines; LOVD, Leiden Open Variation Database; HGMD, Human Gene Mutation Database; VUS, variant of unknown significance; NA, not available; WES, whole-exome sequencing; WGS, whole-genome sequencing.

**Table 2 genes-15-01594-t002:** Candidate variants for patient B detected in the WES dataset. Classification according to American College of Medical Genetics and Genomics (ACMG) guidelines from the Franklin platform.

Gene	cNomen	Zyg.	gnomAD All (%)	ACMG	LOVD	ClinVar	HGMD	Testing Assay
*USH1C*	NM_153676.4:c.1020-2A>C	Het.	0.0001	LP	NC	LP	-	WES
*IFT27*	NM_001177701.3:c.352+1G>T	Het.	0.0059	P	-	P/LP	DM	WES
*ADGRA3*	NM_145290.4:c.2T>C	Het.	0.0001	VUS	-	-	-	WES
*NRL*	NM_006177.3:c.544C>T	Hom.	0.0007	P	-	Conflicting	DM	WES

Abbreviations: cNomen, Human Genome Variation Society (HGVS) cDNA-level nucleotide change nomenclature; gnomAD all (%), genome aggregation database v.4.1.0 overall minor allele frequency in percentage; ACMG, American College of Medical Genetics and Genomics guidelines; LOVD, Leiden Open Variation Database; HGMD, Human Gene Mutation Database; VUS, variant of unknown significance; P, pathogenic; LP, likely pathogenic; DM, disease-causing mutation; NC, not classified; WES, whole-exome sequencing.

**Table 3 genes-15-01594-t003:** Transcript identification and quantification for the *NRL* variant NM_006177.3:c.-41_-28+23del for reference (WT) and variant (MT) minigenes. The table lists the transcripts identified, along with their characteristics, such as length, their relative abundance in reference (WT) and variant (MT) minigenes, the difference (delta) in relative abundance between MT and WT sequencing results, and the effect on the transcript. The table is sorted by relative abundance.

	Transcript	Length	WT (%)	MT (%)	Δ MT-WT (%)	Effect on Transcript
T1	NRL_ex1-NRL_ex2	432 bp	99.6	34.5	−65.1	WT
T2	NRL_ex1del1-NRL_ex2	453 bp	0	23.3	+23.3	altDS_ex1del_1
T3	NRL_ex1del2-NRL_ex2	947 bp	0	10.9	+10.9	altDS_ex1del_2

Abbreviations: WT, wildtype (or reference); MT, mutant (or variant); ex, exon; del, deletion; bp, basepairs; Δ, delta; altDS, alternative donor splice site.

## Data Availability

The original contributions presented in the study are included in the article/Appendix A; further inquiries can be directed to the corresponding author.

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
