# Peer review of "Retinal Dystrophy Associated with Homozygous Variants in *NRL"

_genes, 2024, doi:10.3390/genes15121594_

Round 1
Reviewer 1 Report
Comments and Suggestions for Authors
The manuscript demonstrates significant originality and relevance by expanding the genotypic spectrum of NRL-related retinal dystrophies through the identification of two novel pathogenic variants. The robustness of the methodology and the scientific soundness of the work are supported by the integration of clinical, genetic, and functional approaches, particularly the use of minigene assays to confirm splicing alterations. The content is well-supported by relevant references, and the findings have clear implications for genetic diagnostics and potential therapeutic strategies. While the presentation is generally clear, Figure 5 could be refined to enhance clarity.
Regarding the allele frequency of NM_006177.3.544, I have an observation that could be reviewed by the authors. NM_006177.3.544 corresponds to the identifier rs901811301. According to Gnom AD, there are 1,347,967 C alleles and only 11 T alleles (https://gnomad.broadinstitute.org/variant/14-24081406-G-A?dataset=gnomad_r4). Of these, 2 are found in the Finnish population and 9 in non-Finnish populations. If this information is correct, the authors could discuss its implications for risk genotype predictions in different populations.
Overall, the manuscript represents a valuable contribution to the field of retinal genetics.
Reviewer 2 Report
Comments and Suggestions for Authors
An original research article is a primary scientific publication that presents new research findings, and this paper is a case series. It usually consists of an abstract, introduction, methods, results, discussion, and references. A case report, on the other hand, is a detailed account of a unique clinical or research case, often highlighting novel or rare findings.
It is generally a good practice to spell out acronyms when you first use them. If your readers are unfamiliar with an acronym, they may overlook your paper instead of taking the time to look up its meaning. For example, not everyone knows what "NRL" stands for. However, it’s essential to consider your target audience; there may be certain abbreviations or acronyms that you can assume the readership of the journal will understand.cAnother thing with abbreviations is that you should Avoid beginning a sentence with an acronym or an abbreviation as in ln 98.
The background/ objectives section should be condensed into one sentence.
Materials and methods should be subdivided as follows: 2.1 Demographics, 2.2 Genetics, 2.3 Data analysis.
Reviewer 3 Report
Comments and Suggestions for Authors
This manuscript provides a detailed genetic and clinical characterization of two cases of retinal dystrophy associated with homozygous variants in the NRL gene. It expands the known genotypic spectrum of recessive NRL-related retinal dystrophy and underscores the value of updated bioinformatics workflows and functional assays, such as minigene experiments, for precise variant analysis. The paper is well-organized, and the findings are significant for advancing our understanding of inherited retinal disorders. Below are some comments and suggestions to enhance the clarity and impact of the manuscript:
1. It is recommended to move the subfigure letters (e.g., “(a), (b)”) to the beginning of each sentence in the figure captions. This formatting change will make it easier for readers to refer to the figures while reading the main text.
2. Some explanatory text currently placed in the figure captions should be moved to the main text. For instance, the pigmentary changes observed in Patient A at the age of 16 (Figure 1) are explained in the caption but deserve more discussion in the results or discussion sections.
3. To improve the readability of the ERG data, add labels (e.g., "rod response," "rod-cone response," "light-adapted flicker response," and "single flash response") directly onto the figures themselves rather than embedding them solely in the captions. This will make it easier for readers to interpret the data in conjunction with the text.
4. Some minor questions about the clinical characteristics:
(1) In Figure 1e and 1f, the macula wrinkling observed at age 7 appears absent or diminished at age 16. Is this a confirmed observation, or was it not explicitly assessed at later time points?
(2) Figure 1g indicates non-recordable rod responses at both ages 7 and 11. Why did the patient not report nyctalopia?
(3) How was the b/a wave amplitude ratio of 1.05 calculated from the cone single flash response of Patient B?
5. The authors discussed gene therapy as one of the potential treatment strategies for IRDs. Are there other potential therapeutic methods, such as cell therapy or tissue engineering? Some helpful references:
Lu R, Soden PA, Lee E. Tissue-Engineered Models for Glaucoma Research. Micromachines (Basel). 2020 Jun 24;11(6):612. doi: 10.3390/mi11060612. PMID: 32599818; PMCID: PMC7345325.
Wu A, Lu R, Lee E. Tissue engineering in age-related macular degeneration: a mini-review. J Biol Eng. 2022 May 16;16(1):11. doi: 10.1186/s13036-022-00291-y. PMID: 35578246; PMCID: PMC9109377.
Round 2
Reviewer 2 Report
Comments and Suggestions for Authors
I find this improvement impeccable.